# Does Concurrent Distal Tibiofibular Joint Arthrodesis Affect the Nonunion and Complication Rates of Tibiotalar Arthrodesis?

**DOI:** 10.3390/jcm11123387

**Published:** 2022-06-13

**Authors:** Carsten Schlickewei, Julie A. Neumann, Sinef Yarar-Schlickewei, Helge Riepenhof, Victor Valderrabano, Karl-Heinz Frosch, Alexej Barg

**Affiliations:** 1Department of Trauma and Orthopaedic Surgery, University Medical Center Hamburg-Eppendorf, Martinistr. 52, 20251 Hamburg, Germany; yarar@uke.de (S.Y.-S.); k.frosch@uke.de (K.-H.F.); al.barg@uke.de (A.B.); 2Department of Orthopaedics, University of Utah, 590 Wakara Way, Salt Lake City, UT 84108, USA; julie.neumann@hsc.utah.edu; 3Center for Trauma Rehabilitation and Sport Medicine, BG Hospital Hamburg, Bergedorfer Str. 10, 21033 Hamburg, Germany; helge.riepenhof@redbulls.com; 4Red Bull Athlete Performance Center, Brunnbachweg 71, 5303 Thalgau, Austria; 5Swiss Ortho Center, Schmerzklinik Basel, Swiss Medical Network, Hirschgässlein 15, 4010 Basel, Switzerland; vvalderrabano@swissmedical.net; 6Department of Trauma Surgery, Orthopaedics and Sports Traumatology, BG Hospital Hamburg, Bergedorfer Str. 10, 21033 Hamburg, Germany

**Keywords:** osteoarthritis, ankle fusion, ankle arthrodesis, distal tibiofibular joint, complications, delayed union, nonunion

## Abstract

Tibiotalar arthrodesis successfully treats ankle arthritis but carries risk of nonunion. It is unclear whether concurrent distal tibiofibular arthrodesis affects tibiotalar nonunion rate. The purpose of this study is to compare tibiotalar nonunion and complication rates in patients with versus without a distal tibiofibular arthrodesis. This is a retrospective review of 516 consecutive ankle arthrodesis performed between March 2002 and May 2016. A total of 319 ankles (312 patients) underwent primary, open tibiotalar arthrodesis (227 with distal tibiofibular arthrodesis, 92 without). Primary outcome measure was nonunion rate. Secondary outcome measures were time to tibiotalar union, rate of development of post-operative deep vein thrombosis (DVT)/pulmonary embolism (PE), rate of deep wound complications, and rate of return to operating room (OR). No differences in nonunion rates were observed in both cohorts of patients with versus without distal tibiofibular arthrodesis: 17/227 (7.5%) versus 11/92 (12%) (*p* = 0.2), respectively, odds ratio was 0.74, 95% CI: 0.29~2.08 (*p* = 0.55). There was no difference in deep wound complications (5.3% versus 10.9%, *p* = 0.42), time to union (3.7 months versus 4.1 months, *p* = 0.72), or rate of development of DVT/PE (5.2% versus 2.2%, *p* = 0.18) between patients with and without distal tibiofibular arthrodesis, respectively. This is the first study directly comparing nonunion and complication rates in primary, open ankle arthrodesis with and without distal tibiofibular arthrodesis. Inclusion of the distal fibular joint with the tibiotalar fusion was not associated with a change in tibiotalar nonunion rate, time to union, wound complications, or postoperative DVT/PE.

## 1. Introduction

Approximately 1% of the world’s population is affected with ankle osteoarthritis [1]. The posttraumatic etiology is the most common etiology of the end-stage ankle osteoarthritis [1,2,3,4,5,6,7]. After failure of nonoperative treatments for severe ankle arthritis [5] including bracing/immobilization, anti-inflammatories, shoe wear and activity modification, and corticosteroid injections, patients have two operative options—arthroplasty [8,9,10,11,12] and arthrodesis [13,14,15,16,17,18]. Tibiotalar arthrodesis still remains the current gold standard [19]. Ankle arthrodesis is generally reliable procedure in terms of postoperative pain relief and deformity correction [20], but as with any surgery there can be complications [21]. One of the most serious complications are delayed union or nonunion at the tibiotalar side [22,23,24]. Unfortunately, nonunions often results in a secondary surgery [22,25]. Several risk factor for nonunion have been identified including osteonecrosis of talus, smoking, poor bone quality, inherent positional ankle deformity, infection, major medical problems, and prior open injury, to name a few [21,22,24].

Over the years, in an attempt to improve the rate of union, ankle arthrodesis has been performed with a wide variety of fixation methods and operative techniques [26]. No matter the technique, the goal is to achieve a pain free, plantigrade foot [27]. Goldthwait from Boston was the first surgeon to describe the transfibular approach for ankle arthrodesis in patients with infantile paralysis [28]. Horwitz described this method in 1942 in more detail [29]. Since then several reports demonstrated reliable postoperative results following this procedure [30]. There are advantages and disadvantages of a transfibular technique that includes onlay grafting of the lateral malleolus to the distal tibia when compared to an anterior, posterior, or medial approach. Advantages include improved wound healing, lower procedural cost due to less expensive screws when compared to plates [31], use of localized fibular graft instead of other-site donor site morbidity [32,33,34], clear exposure of joint surfaces of both tibia and talus [30,31,35], the ability to use a stable, scaffolding fibular strut graft with a posterior soft tissue flap thus its own vascularity [30], another bony surface (medial side of fibula) for fusion [30,36], inhibition of valgus drift in cases of delayed union [36], and decreased risk to the superficial peroneal and sural nerves [35]. Disadvantages associated with transfibular approach include another joint that may go on to nonunion (distal tibiofibular joint), persistent pain at the fibular osteotomy site, difficulty with converting a tibiotalar fusion to a total ankle arthroplasty [37,38], and destabilization of the ankle theoretically increasing tibiotalar nonunion rate [36].

The purpose of this study is to evaluate whether or not the addition of a distal tibiofibular arthrodesis affects the nonunion rate in patients undergoing primary open ankle arthrodesis. The hypothesis was that the addition of a distal tibiofibular arthrodesis would decrease the nonunion rate in patients undergoing primary open ankle arthrodesis due to added stability of the construct.

## 2. Materials and Methods

### 2.1. Study Participants

Between March 2002 and May 2016, 516 consecutive patients underwent tibiotalar arthrodesis. These patients were identified by CPT codes followed by a chart review. After retrospective review, 319 ankles from 312 patients met inclusion criteria. Inclusion criteria were failure of a minimum of 3 months of non-operative treatment [5], tibiotalar arthritis (Takakura Stage 2, 3 or 4) [39], and a minimum of 6 months follow-up (Figure 1 and Figure 2). Exclusion criteria were previous hindfoot surgery, arthroscopic tibiotalar arthrodesis, active septic arthritis, or age < 18-years-old. All patients in this study underwent primary open tibiotalar arthrodesis by one of four fellowship-trained orthopedic foot and ankle surgeons. A total of 227 of these patients underwent open tibiotalar arthrodesis with a distal tibiofibular arthrodesis and 92 patients underwent open tibiotalar arthrodesis without a distal tibiofibular arthrodesis. This study was conducted in accordance with the Declaration of Helsinki and the Guidelines for Good Clinical Practice. The data of this study were retrospectively and fully anonymized evaluated.

### 2.2. Surgical Technique

Operative approach, fixation method, choice of allograft/autograft/bone graft substitutes were varied based on surgeon and patient factors. Figure 2 shows radiographs of a typical case in this study in which the distal tibiofibular arthrodesis was included in the construct. In each case the foot was positioned in slight valgus, external rotation, and dorsiflexion and provisionally fixated for fluoroscopic imaging. If the foot position was deemed satisfactory, the tibiotalar joint +/− distal tibiofibular joint were fixated. Final fluoroscopic images were routinely obtained prior to closing the wounds and placing post-operative immobilization with use of a splint or boot. There was not a standardized post-operative protocol, but all patients were instructed to be non-weight bearing for a minimum of 6 weeks after surgery. Generally, patients were followed at 2, 6, and 12 weeks, 6 month, and 1-year intervals.

### 2.3. Outcomes Analysis

The primary outcome measure was nonunion rate of tibiotalar arthrodesis [2]. Union was defined based on patient reported symptoms and clinical physical examination criteria (no pain, no warmth, improvement in swelling, and stability to stress) and radiographic criteria (visible trabecular bridging across the arthrodesis site and no lucency around the hardware) [40]. Appropriate osseous union was defined as trabecular bridging across the tibiotalar joint (at least 80%) within 6 postoperative months [41] (Figure 3 and Figure 4). In patients with nonunion, computerized tomography (CT) was utilized for further assessment and was evaluated by independent radiologist not involved in any of surgeries. Secondary outcome measures were wound complications, return to the operating room, and rate of development of post-operative deep vein thrombosis (DVT), or pulmonary embolism (PE).

### 2.4. Statistical Analysis

Patient demographics and clinical characteristics were summarized descriptively for all ankle arthrodesis cases and also stratified by with or without distal tibiofibular arthrodesis. Continuous variables that were approximately normally distributed were summarized as mean and standard deviation (SD) and compared between the two groups using a *t*-test. Variables that were not normally distributed were summarized as median and interquartile range (IQR) and compared using Wilcoxon rank sum test. Categorical variables were summarized as frequency and percent and compared using a chi-squared test or Fisher’s exact test. Nonunion, wound complications, and DVT/PE were summarized as frequency and percentage and compared between groups using chi-squared or Fisher’s exact test. Months to union for ankles that had achieved union were summarized as median (IQR) and compared using a Wilcoxon rank sum test. Because rate of return to OR depended on patient follow-up, it was summarized as rate per person-year and a log rank test was used to compare with and without distal tibiofibular arthrodesis groups. The primary outcome, nonunion was further compared with the distal tibiofibular arthrodesis group using logistic regression adjusting for age, sex, year of surgery, smoking status, and BMP use. Odds ratio with 95% confidence interval were reported for all covariates. Statistical significance was defined as *p* < 0.05, and all tests were two-sided. All data were analyzed using IBM SPSS Statistics version 26.0 (IBM, Armonk, NY, USA).

## 3. Results

### 3.1. Subjects

Average age of the patients was 57.7 ± 13.5 years (range 18–88.8). Median follow-up time was 27.5 months (range 17–43). Demographic and clinical characteristics are shown in Table 1. Patients without a distal tibiofibular arthrodesis were significantly younger, but otherwise all demographic and clinical characteristics were comparable. Preoperative deformity measures in patients with and without distal tibiofibular arthrodesis were compared in Table 2, and there were no significant differences. Etiology of ankle arthritis is shown in Table 3. Again, no significant differences were seen between both cohorts. Comparison of the fixation type, operative approach, and bone graft details are demonstrated in the Table 4, Table 5 and Table 6, respectively.

### 3.2. Postoperative Outcomes/Complications

Outcomes by procedure type are shown in Table 7. There was no significant difference between the nonunion rate of patients who underwent distal tibiofibular arthrodesis (17/227 (7.5%)) and the nonunion rate of patients who did not undergo distal tibiofibular arthrodesis (11/92 (12%)) (*p* = 0.2). There was also no significant difference in the time to union, rate of deep wound complications, and rate of DVT/PE between the cohorts. Table 8 shows the rate of return to the OR. Table 9 shows the reason of return to the OR between the cohorts. Table 10 shows multivariable logistic regression predicting nonunion.

## 4. Discussion

This study shows that inclusion of the distal tibiofibular arthrodesis in primary open tibiotalar arthrodesis does not affect immediate complication rates including nonunion, wound complications, time to union, rate of return to the OR, and DVT/PE rate. 

Patients who had a distal tibiofibular arthrodesis had autograft 208/227 (91.6%) of the time versus 64/92 (69.6%) of those who did not (*p* < 0.001). This is because autograft from the distal fibula was easily harvested for inclusion in the fusion mass. Not surprisingly, patients who have a distal tibiofibular arthrodesis were less likely to have utilized allograft 37/227 (16.3%) versus 23/92 (25%) of those without.

A relatively high number of 8/319 (2.5%) patients included in this study ultimately underwent a transtibial amputation (4 with distal tibiofibular arthrodesis, 4 without). Three of the four in the distal tibiofibular arthrodesis group had the tibiotalar arthrodesis for Charcot neuroarthropathy and the fourth patient underwent an amputation for an infected tibiotalar arthrodesis nonunion after prior equinovarus deformity correction. In the group without distal tibiofibular arthrodesis, one patient had a tibiotalar nonunion after attempted ankle arthrodesis after a failed total ankle arthroplasty. The second patient had a tibiotalar arthrodesis for Charcot neuroarthropathy. The third patient had chronic pain and deformity after multiple surgeries for clubfoot. The fourth patient had an opened pilon that had failed multiple open reduction, internal fixations and finally the ankle fusion with a blade plate. 

The authors recognize that approximately 70% of patients suffering from the severe pain and disability associated with ankle arthritis are from a post-traumatic cause [21,36,42]. This means that patients often have previous incisions and possibly an impaired periarticular soft tissue envelope, which should be considered when planning an ankle arthrodesis. The results of this study suggest that surgeons can approach tibiotalar fusion from the approach that makes the most sense, as the inclusion of distal tibiofibular arthrodesis was not associated with significant changes in complication rates or time to fusion.

There are a number of case series describing outcomes after a transfibular approach for tibiotalar arthrodesis, but none of them directly compares the transfibular approach with an anterior approach. In 1998, Mann et al. reported an 88% fusion rate in a single surgeon series using a transfibular approach on 81 ankles at a minimum follow-up of 12 months [43]. Napiontek et al. report 23 ankles that underwent a transfibular approach, and only 1 of the 23 had a nonunion at an average of 32 months follow-up [31]. In 2007, Colman et al. evaluated 48 ankles at an average of 45 months post-operatively and showed a 96% union rate in ankles arthrodesis through a transfibular approach [30]. This study did not compare subjects to other approaches [30]. In a retrospective study on 29 patients from 2009 and 2014, Balaji et al. reported a 100% union rate at a mean of 3.8 months post-operatively where they utilized a transfibular approach with the fibula as an onlay graft [44]. They reported four superficial infections, no deep infections and they reported that patients with post-traumatic had better functional outcomes than those with rheumatoid or tuberculosis arthritis [44]. In 2016, Lee et al. reported 1 nonunion in 23 cases (92% union rate) in a heavy smoker utilizing a single lateral incision [32]. Recently, Park et al. demonstrated promising short-term results in a patient cohort consisting of 35 persons with a mean age of 66.5 years [17]. In this publication, an anterior approach was utilized in combination with transfibular approach in order to completely debride the joint surfaces including medial gutter and to increase the initial stability of the osteosynthetic fixation using anterior plate. Union was achieved in all cases with statistically significant pain relief from 6.8 (6–9) to 2.6 (0–7) using VAS [17]. Based on our experience, we routinely perform medial arthrotomy of the tibiotalar joint using separate incision to visualize and to debride the medial gutter [45]. In 2020, Suo et al. evaluated the clinical outcome of ankle arthrodesis with screw fixation through the transfibular approach for end-stage ankle arthritis in 28 ankles using a novel internal fixation screw [46]. The headless compression hollow screw was developed to have the mechanical properties of a taper, a full thread, and a change in thread pitch to better stabilize the joint complex and promote bone healing. Although the study showed satisfactory clinical results, it did not include a comparison with other methods of internal fixation.

This study has several limitations. The operative technique, fixation hardware, as well as post-operative protocols were not standardized as patients in this series were treated by one of four foot and ankle fellowship trained surgeons. Specifically, 192/227 (84.6%) of patients who had a distal tibiofibular arthrodesis had their tibiotalar fixation with screws alone versus the majority of patients 47/92 (51.1%) who did not have their distal tibiofibular arthrodesis included had a plate as their main fixation method. Therefore, one may argue that this is a study comparing nonunion rate between fixation method rather than inclusion of distal tibiofibular arthrodesis. The same could be said about the operative approach. The majority of patients 182/227 (80.2%) who had their distal tibiofibular arthrodesis included in the tibiotalar fusion had a lateral approach whereas most of the patients who had no distal tibiofibular arthrodesis had an anterior 47/92 (51.1%) or posterior 11/92 (12%) approach. One flaw in this study is neglect to account for concomitant surgeries, many of which may attribute to rate of nonunion. This study did not examine the impact of preexisting risk factors of nontraumatic ankle arthritis, such as those associated with rheumatoid arthritis, hemochromatosis, hemophilia, or Charcot neuroarthropathy [6]. These conditions present different challenges and may have different negative effects on postoperative outcomes and complications. Additionally, there is human error associated with performing chart review thus some data may be incomplete or incorrect. Lastly, in this study, tibiotalar union was assessed using conventional weight bearing radiographs. It has been demonstrated, that CT scans have higher reliability in determining the successful fusion of the hindfoot [47]. However, in our study only patients with suspected nonunion were selected for CT scan, as it would be impractical to obtain CT scans for all patients postoperatively.

## 5. Conclusions

This is the first study directly comparing nonunion and complication rates in primary, open ankle arthrodesis with and without distal tibiofibular arthrodesis. In this study, inclusion of the distal tibiofibular arthrodesis in tibiotalar arthrodesis fusion mass does not affect nonunion rate nor does it affect rate of wound complication, time to union, and DVT/PE rate.

## Figures and Tables

**Figure 1 jcm-11-03387-f001:**
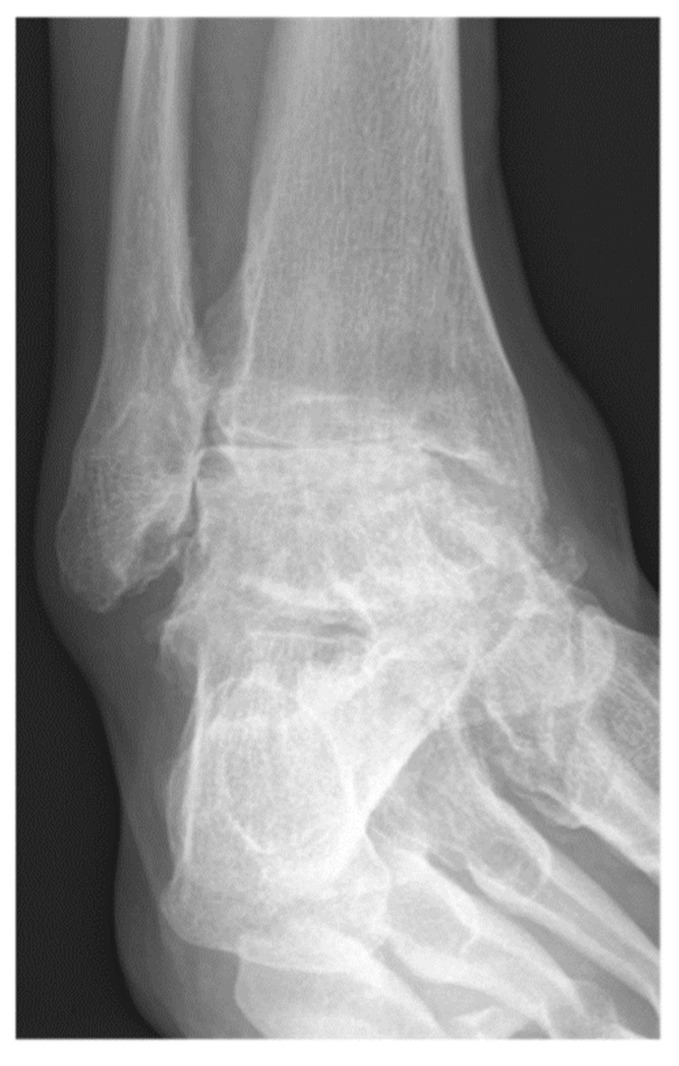
A 57-year-old male patient presenting end-stage varus tibiotalar osteoarthritis.

**Figure 2 jcm-11-03387-f002:**
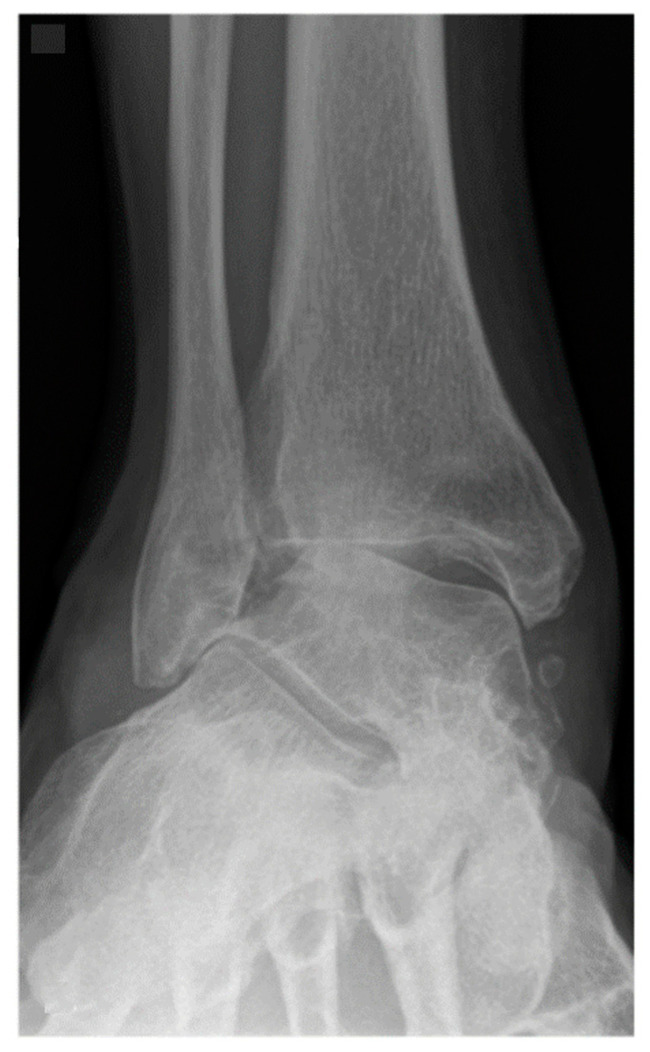
A 62-year-old female patient presenting end-stage valgus tibiotalar osteoarthritis.

**Figure 3 jcm-11-03387-f003:**
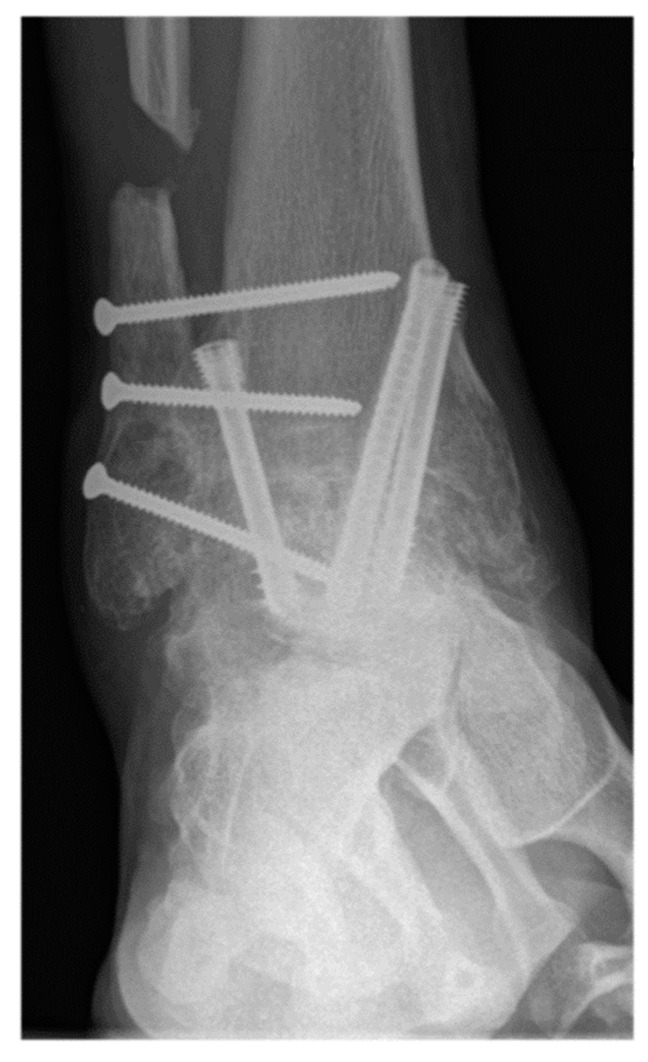
A 57-year-old male patient presenting complete union of the tibiotar and distal tibiofibular joints at 6 months follow-up.

**Figure 4 jcm-11-03387-f004:**
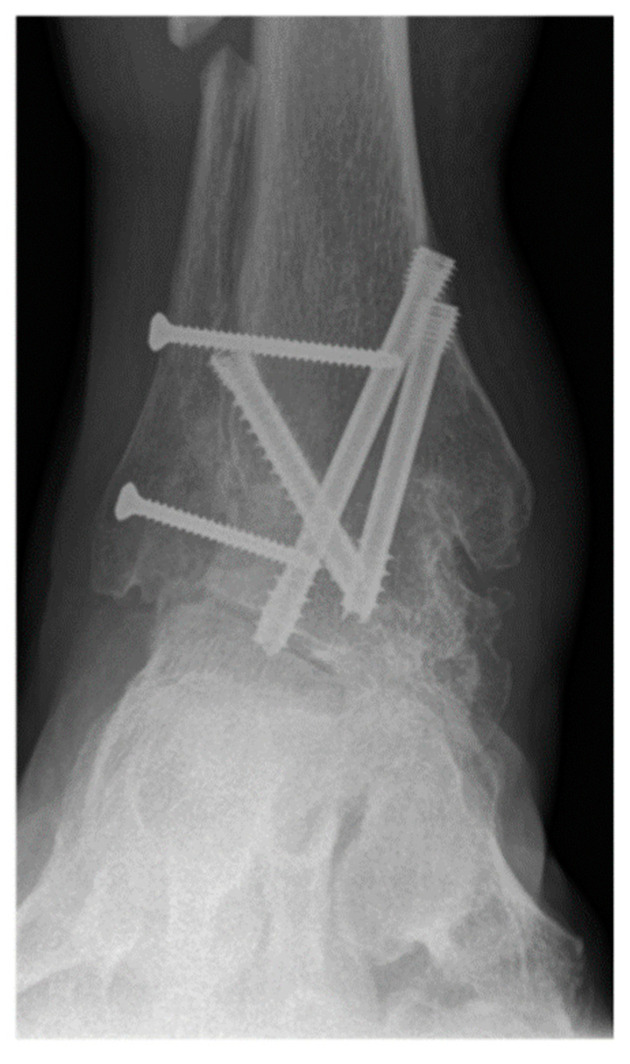
A 62-year-old female patient presenting complete union of the tibiotar and distal tibiofibular joints at 4 months follow-up.

**Table 1 jcm-11-03387-t001:** Demographic and clinical characteristics.

Variable	With Tibiofibular Arthrodesis (*n* = 227)	Without Tibiofibular Arthrodesis (*n* = 92)	All Patients (*n* = 319)	*p*-Value
Age [years ± SD]	58.8 ± 12.4	54.9 ± 15.7	57.7 ± 13.5	***0.036*** ^†^
Gender (male)	133 (58.6%)	51 (55.4%)	184 (57.7%)	0.61 ^‡^
BMI with range [kg/m^2^]	29.2 (25.8–33.1)	301 (26.4–34.2)	29.4 (25.9–33.8)	0.32 ˆ
Diabetes	35 (15.4%)	16 (17.4%)	51 (16%)	0.66 ^‡^
Smokers	34 (15%)	13 (14.1%)	47 (14.7%)	0.85 ^‡^
Right sided surgery	117 (51.5%)	51 (55.4%)	168 (52.7%)	0.53 ^‡^

^†^*t*-test; ^‡^ Chi-squared test; ˆ Wilcoxon rank sum test. BMI = Body Mass Index; SD = Standard deviation.

**Table 2 jcm-11-03387-t002:** Pre-operative deformity measures.

Pre-Operative Deformity	With Tibiofibular Arthrodesis (*n* = 227)	Without Tibiofibular Arthrodesis (*n* = 92)	All Patients (*n* = 319)	*p*-Value
MDTA with range [°]	88° (86–91°)	88° (85–90°)	88° (85–91°)	0.15 ^†^
TTT with range [°]	0° (−5–3°)	0° (−2–3°)	0° (−4–3°)	0.21 ^†^
CMA with range [°]	−5.1° (−19.3–12.9°)	−6.4° (−15.8–10.2°)	−5.8° (−17.5–11.9°)	0.98 ^†^
ADTA with range [°]	82° (79–87°)	82° (77–86°)	82° (78–86.8°)	0.18 ^†^

^†^ Wilcoxon rank sum test. ADTA = anterior distal tibial angle; CMA = calcaneal moment arm (negative values indicated varus malalignment); MDTA = medial distal tibial angle; TTT = tibiotalar test.

**Table 3 jcm-11-03387-t003:** Etiology of tibio-talar arthritis.

Etiology	With Tibiofibular Arthrodesis (*n* = 227)	Without Tibiofibular Arthrodesis (*n* = 92)	All Patients (*n* = 319)	*p*-Value
Primary	7 (3.1%)	5 (5.4%)	12 (3.8%)	
Secondary (including posttraumatic OA)	220 (96.9%)	87 (94.6%)	307 (96.2%)	0.34 ^†^

^†^ Fisher’s exact test. OA = osteoarthritis.

**Table 4 jcm-11-03387-t004:** Type of fixation.

Type of Fixation	With Tibiofibular Arthrodesis (*n* = 227)	Without Tibiofibular Arthrodesis (*n* = 92)	All Patients (*n* = 319)	*p*-Value
Blade plate	5 (2.2%)	11 (12.0%)	16 (5.0%)	** * <0.001 * ** ^†^
External fixation alone	4 (1.8%)	13 (14.1%)	17 (5.3%)	
External fixation + plate	1 (0.4%)	0 (0%)	1 (0.3%)	
IM nail	14 (6.2%)	5 (5.4%)	19 (6%)	
Plate	11 (4.8%)	47 (51.1%)	58 (18.2%)	
Screws	192 (84.6%)	16 (17.4%)	208 (65.2%)	

^†^ Fisher’s exact test. IM = intramedullary.

**Table 5 jcm-11-03387-t005:** Operative approach.

Operative Approach	With Tibiofibular Arthrodesis (*n* = 227)	Without Tibiofibular Arthrodesis (*n* = 92)	All Patients (*n* = 319)	*p*-Value
Anterior	8 (3.5%)	47 (51.1%)	55 (17.2%)	** * <0.001 * ** ^†^
Lateral	182 (80.2%)	20 (21.7%)	202 (63.3%)	
Medial	0 (0%)	3 (3.3%)	3 (0.9%)	
Medial and anterior	0 (0%)	1 (1.1%)	1 (0.3%)	
Medial and lateral	31 (13.7%)	10 (10.9%)	41 (12.9%)	
Posterior	6 (2.6%)	11 (12%)	17 (5.3%)	

^†^ Fisher’s exact test.

**Table 6 jcm-11-03387-t006:** Bone graft details.

Autograft or Allograft	Type	With Tibiofibular Arthrodesis (*n* = 227)	Without Tibiofibular Arthrodesis (*n* = 92)	All Patients (*n* = 319)	*p*-Value
Autograft	Yes	208 (91.6%)	64 (69.6%)	272 (85.3%)	** * <0.001 * ** ^†^
	Distal fibula ± distal tibia	159 (70%)	21 (22.8%)	180 (56.4%)	
	Iliac crest	5 (2.2%)	7 (7.6%)	12 (3.8%)	
	Proximal tibia	44 (19.4%)	35 (38%)	79 (24.8%)	
	Unspecified	0 (0%)	1 (1.1%)	1 (0.3%)	
Allograft	Yes	37 (16.3%)	23 (25%)	60 (18.8%)	0.07 ^‡^
	Bone block	6 (2.6%)	7 (7.6%)	13 (4.1%)	
	Cancellous chips	25 (11%)	11 (12%)	36 (11.3%)	
	Cortical bone powder	0 (0%)	1 (1.1%)	1 (0.3%)	
	DBM	6 (2.6%)	4 (4.3%)	10 (3.1%)	0.48 ^†^
	BMP	20 (8.8%)	20 (21.7%)	40 (12.5%)	** * 0.002 * ** ^‡^

^†^ Fisher’s exact test; ^‡^ Chi-squared test. BMP = bone morphogenic protein; DBM = demineralized bone matrix.

**Table 7 jcm-11-03387-t007:** Outcomes.

Variable	With Tibiofibular Arthrodesis (*n* = 227)	Without Tibiofibular Arthrodesis (*n* = 92)	All Patients (*n* = 319)	*p*-Value
Nonunion	17 (7.5%)	11 (12%)	28 (8.8%)	0.20 ^†^
Months to union	3.5 (1.4–18.2)	4.1 (1.6–13.5)	3.8 (2.9–6.2)	0.72 ^‡^
Wound complications Superficial Deep	59 (26%) 47 (20.7%) 12 (5.3%)	28 (30.4%) 18 (19.6%) 10 (10.9%)	87 (27.3%) 65 (20.4%) 22 (6.9%)	0.42 ^‡^
Thrombosis DVT PE	11 (4.8%) 1 (0.4%)	1 (1.1%) 1 (1.1%)	12 (3.8%) 2 (0.6%)	0.18 *

^†^ Chi-squared; ^‡^ Wilcoxon rank sum test; * Fisher’s exact test; DVT = deep venous thrombosis; PE = pulmonary embolism.

**Table 8 jcm-11-03387-t008:** Rate of return to OR by procedure type excluding planned removal of external fixation.

Rate of Return to OR	With Tibiofibular Arthrodesis (*n* = 227)	Without Tibiofibular Arthrodesis (*n* = 92)	All Patients (*n* = 319)	*p*-Value
Rate (#/yr)	55 (24.2%) 0.044/person year	20 (21.7%) 0.056/ person year	86 (27%) 0.053/person year	0.89 ^†^

^†^ Log-rank test. OR = operating room.

**Table 9 jcm-11-03387-t009:** Reason for return to OR by procedure type, excluding planned removal of external fixation.

Reason for Return to OR	With Tibiofibular Arthrodesis (*n* = 227)	Without Tibiofibular Arthrodesis (*n* = 92)	All Patients (*n* = 319)	*p*-Value
Amputation	4 (1.8%)	4 (4.3%)	8 (2.5%)	0.12 ^†^
I&D (operative site)	7 (3.1%)	4 (4.3%)	11 (3.4%)	
I&D (autograft site)	1 (0.4%)	0 (0%)	1 (0.3%)	
I&D + ROH	1 (0.4%)	0 (0%)	1 (0.3%)	
Osteotomy	1 (0.4%)	1 (1.1%)	2 (0.6%)	
Relocation of peroneus brevis tendon	1 (0.4%)	1 (1.1%)	2 (0.6%)	
Revision ankle arthrodesis	14 (6.2%)	3 (3.3%)	17 (5.3%)	
ROH	20 (8.8%)	7 (7.6%)	26 (8.2%)	
Soft tissue reconstruction	2 (0.9%)	0 (0%)	2 (0.6%)	
Subtalar arthrodesis	3 (1.3%)	0 (0%)	3 (0.9%)	

^†^ Fisher’s exact test. I&D = irrigation and debridement, includes hematoma evacuation and I&D for infection OR = operating room; ROH = removal of hardware.

**Table 10 jcm-11-03387-t010:** Multivariable logistic regression predicting nonunion.

Variables	Odds Ratio	*p*-Value
Inclusion of distal fibular strut autograft in tibiotalar fusion	0.74 (0.29~2.08)	0.55
Year of surgery	0.94 (0.84~1.05)	0.27
Age at time of surgery	1.02 (0.99~1.06)	0.29
Male gender	1.2 3(0.52~3.00)	0.64
Smoker	0.90 (0.20~2.99)	0.88
+BMP	0.60 (0.09~2.37)	0.53
Pre-operative MDTA	1.03 (0.97~1.10)	0.33

BMP = bone morphogenic protein; MDTA = medial distal tibial angle.

## Data Availability

Derived data supporting the findings of this study are available from the author upon reasonable request.

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
