# Peer review of "Does Concurrent Distal Tibiofibular Joint Arthrodesis Affect the Nonunion and Complication Rates of Tibiotalar Arthrodesis?"

_jcm, 2022, doi:10.3390/jcm11123387_

Round 1

Reviewer 1 Report

I have no complaints of the method or the writing as such. The manuscript is clear and complete also in regards to the  bibliographical references, but  I suggest you mention this bibliographic reference:

Suo H, Fu L, Liang H, Wang Z, Men J, Feng W. Orthop Surg. 2020 Aug;12(4):1108-1119.End-stage Ankle Arthritis Treated by Ankle Arthrodesis with Screw Fixation Through the Transfibular Approach: A Retrospective Analysis. Orthop Surg. 2020 Aug;12(4):1108-1119.

In this study, “ two or three 7.3 × 60 mm headless compression full‐threaded cannulated screws were used to cross‐fix the tibiotalar joint, and three or four 3.5 × 34 mm cortical screws were used to fix the fibula to the tibia and talus. As a new type of internal fixation screw, a headless compression hollow screw is designed to have the mechanical characteristics of taper, full thread, and thread pitch change, which can continuously compress the tibia and talus, promoting the healing of bone and enhancing the stability of the joint complex. In addition, the headless design eliminates the need for conventional countersunk head treatment. Hollow screws are more resistant to bending than cancellous screws.” 

Author Response

Dear Reviewer,

thank you very much for your review.

We have taken up your recommendation and added the bibliographic reference in the discussion section.

In 2020 Suo et al. evaluated the clinical outcome of ankle arthrodesis with screw fixation through the transfibular approach for end-stage ankle arthritis in 28 ankles using a novel internal fixation screw [46]. The headless compression hollow screw was developed to have the mechanical properties of a taper, a full thread, and a change in thread pitch to better stabilize the joint complex and promote bone healing. Although the study showed satisfactory clinical results, it did not include a comparison with other methods of internal fixation.

  1. Suo, H.; Fu, L.; Liang, H.; Wang, Z.; Men, J.; Feng, W. End-stage Ankle Arthritis Treated by Ankle Arthrodesis with Screw Fixation Through the Transfibular Approach: A Retrospective Analysis. Orthopaedic Surgery 2020,12,1108-1119, doi: 10.1111/os.12707.

Thank you and best regards

Carsten Schlickewei

Reviewer 2 Report

Article, clear and well structured, is really interesting, full of meaningful content and appropriate references. The experimental design is appropriate to test the hypothesis and conclusions are coherent to obtained results. As shown by the authors, most of patients who developed complications had Charcot arthropathy; therefore we ask to better explain why, according to the authors and literature (for more contents 10.4081/or.2020.8670), this deformity favors, more than other risk factors, complications genesis.

Author Response

Dear Reviewer,

I have taken up your recommendations and discussed your comments in detail with the other authors.

You are certainly correct in your objection.

Unfortunately, we did not investigate the additional risk factors associated with nontraumatic ankle osteoarthritis, as in Charcot neuroarthropathy (advanced age, long-standing diabetes, high hyperglycemia, neuropathy, PAOD).

Therefore, we cannot make a valid statement on whether Charcot neuropathy in particular promotes the development of complications more than other risk factors.

However, we of course understand your objection and have therefore included this fact as a limitation of the paper. I hope you agree with this.

This study did not examine the impact of preexisting risk factors of nontraumatic ankle arthritis, such as those associated with rheumatoid arthritis, hemochromatosis, hemophilia, or Charcot neuroarthropathy [6]. These conditions present different challenges and may have different negative effects on postoperative outcomes and complications.

With best regards

Carsten Schlickewei